# Differential Treatment Responses in Pakistani Schizophrenia Samples: Correlation with Sociodemographic Parameters, Drug Addiction, Attitude to the Treatment and Antipsychotic Agents

**DOI:** 10.3390/brainsci13030407

**Published:** 2023-02-26

**Authors:** Umme Habiba, Aafia Malik, Ghazala Kaukab Raja, Muhammad Raza Memon, Asad Tameezud din Nizami, Rafaqat Ishaq, Muhammad Ilyas, Hadi Valadi, Muhammad Nawaz, Pakeeza Arzoo Shaiq

**Affiliations:** 1University Institute of Biochemistry and Biotechnology, Pir Mehr Ali Shah, Arid Agriculture University Rawalpindi, Shamsabad, Rawalpindi 46300, Pakistan; 2Department of Psychiatry, Jinnah Hospital Usmani Road, Quaid-i-Azam Campus, Lahore 54550, Pakistan; 3Department of Psychiatry, Liaquat University of Medical and Health Sciences, Jamshoro 76090, Pakistan; 4Institute of Psychiatry, WHO Collaborating Center for Mental Health, Benazir Bhutto Hospital, Murree Road, Rawalpindi 23000, Pakistan; 5Department of Rheumatology and Inflammation Research, Institute of Medicine, Sahlgrenska Academy, University of Gothenburg, 41346 Gothenburg, Sweden

**Keywords:** schizophrenia, hallucinations, medication, antipsychotics

## Abstract

Schizophrenia patients demonstrate variations in response to different therapies that are currently being used for the treatment of disorders, such as augmentation therapy (ECT or mood stabilizer) and combination therapy (with antipsychotics). These therapies are also used to treat schizophrenia patients in Pakistan; however, patients show poor overall response. Therefore, this study was conducted to investigate the association between the patients’ response to treatment and the use of antipsychotic agents, with variability in overall response, within different groups of patients. **Methods:** We conducted a retrospective study that included schizophrenia subjects (N = 200) belonging to different age groups, ethnicities, and regions from different outpatient and inpatient departments in psychiatric institutes located in different cities of Pakistan. These patients were assessed for their response to treatment therapies and categorized into four groups (non-responders (N-R), slow response (S-R), patients with relapse, and completely recovered patients (C-R)) according to their responses. **Results:** The final analysis included 200 subjects, of which 73.5% were males. Mean age was 34 ± 10 years. Percentage of N-R was 5%, S-R was 42%, patients with relapse were 24%, and C-R was 1.5%. The generalized linear regression model shows a significant association between medication response and age (*p* = 0.0231), age of onset (*p* = 0.0086), gender (*p* = 0.005), and marital status (*p* = 0.00169). Variability within the medication responses was a result of the treatment regime followed. Antipsychotic agents were significantly associated with the treatment response (*p* = 0.00258, F = 4.981) of the patients. Significant variation was also observed in the treatment response (*p* = 0.00128) of the patients that were given augmentation therapy as well as combination therapy. **Conclusion:** The data suggests proper monitoring of patients’ behavior in response to treatment therapies to implement tailored interventions. Despite several genetic studies supporting the heritability of schizophrenia, an insignificant association between characteristic features and family history might have been due to the limited sample size, suggesting collaborative work with massive sample sizes.

## 1. Introduction

Schizophrenia is complex neuropsychiatric disorder characterized by severe psychotic symptoms affecting the victims and their families in a devastating way. The exact prevalence of schizophrenia in Pakistan is not yet known; however, it is estimated at between 1 and 2% [1]. The risk of schizophrenia increases when there is a family history of any psychiatric disorder, such as bipolar disorder [2]. Generally, a relapse in schizophrenia patients is observed, with an increase in the symptoms’ severity [3]. Individual twin studies and meta-analysis studies have estimated heritability for schizophrenia to be approximately 80%. It is also observed that medical help is not generally sought for schizophrenia patients in developing countries such as Pakistan, which further complicates efforts to correctly calculate its prevalence [4]. The onset of schizophrenia occurs in the early twenties or late adolescence and follows a recurrent and chronic course [5]. Many epidemiological studies have considered the impact of environment on the development of schizophrenia [6]. Certain childhood and adolescent risk factors predict the age of onset of psychosis in patients [7]. Various environmental risk factors that have previously been studied are obstetric complications, winter or spring birth, birth complications, substance abuse, etc. [8]. Similarly, substance abuse disorders are also common among schizophrenia patients. The use of cocaine, alcohol, tobacco and cannabis is common and, it might be due to the increased vulnerability of schizophrenia patients to drug addiction due to genetic determinants. It could also serve as an additional risk factor for the appearance of psychotic symptoms in schizophrenia [9].

The diagnosis of schizophrenia is based on positive and negative symptoms (PANSs) and cognitive impairment. Sometimes, cognitive deficits are considered as related to negative symptoms [10]. Mainly, the negative symptoms are amotivation, reduced expression, social withdrawal, and poor relationships, and the positive symptoms include delusions, hallucinations, and disorganized speech, thinking and behavior [11]. According to DSM-IV, five characteristic symptoms for the diagnosis of schizophrenia, with the requirement that at least two of the symptoms are hallucinations and delusions, must be present for about 1 month [12]. Clinical symptoms of schizophrenia are complicated due to poor or no compliance for the disorder in the early stages, so the patients do not recover completely, and they are at higher risk of relapse after recovery [13]. For the medication therapy of schizophrenia, neuroleptics are frequently used and are usually of two types, typical anti-psychotics and atypical antipsychotics. The main difference between the typical and atypical antipsychotics lies in their mode of action. Typical antipsychotics act on the dopaminergic system, blocking dopamine type 2 (D2) receptors, while atypical antipsychotics act on the dopaminergic and serotoninergic receptors [14]. Therefore, atypical antipsychotic drugs are more effective across a broader range of symptoms of schizophrenia than typical antipsychotic drugs. They are also dramatically less likely to cause the extrapyramidal and endocrine effects that greatly impair quality of life; therefore, they are frequently used [15]. Antipsychotic medication is effective in reducing acute symptoms [16], and long-term antipsychotic medication is recommended for people with recurrent episodes on the basis of evidence that antipsychotic medication reduces the risk of relapse compared to discontinuation [17]. The adverse effects of antipsychotics range from minor tolerability issues to those life threatening. Some adverse effects have little short-term clinical implications but may involve the long-term risk of medical complications [18]. Some side effects associated with antipsychotics and other medications also include dystonia, malignant syndrome, parkinsonism, tardive dyskinesia, and akathisia [19]. Due to the side effects associated with the use of antipsychotics, their effectiveness is found to be limited [18]. There are no reports concerning the prevention of schizophrenia progression by effective treatment in combination with neuroleptics because, mostly, the treatment has less effect against symptoms. A number of factors that affect the compliance, have been evaluated worldwide. These include the patients’ belief in their illness and the benefits of treatment therapies that patients can achieve [20,21]. The estimated time lapse between disease onset and its treatment ranges from 22 weeks to almost 150 weeks [22].

The most accepted definition of response in schizophrenia research is ≥50% improvement on a respective rating scale from baseline to endpoint. Response from the patients to the medication is usually variable, though most patients with first-episode psychosis respond very well to antipsychotic medication, with estimates ranging as high as 87%, but it also includes patients with partial or complete non-response [23]. The efficacy of the treatments devised for schizophrenia is negligible for about one third of the schizophrenia patients, which are termed non-responders [24]. Usually, the treatment course and prognosis are roughly divided into three categories. In one (approximately 25% of the patients),a full response to treatment is usually shown that leads to recovery from the first episode; the other one (around 50% of patients) shows recurrent illness in the form of exacerbations and remissions; and the third groups of patients (25% of patients), show an unfavorable course and incomplete response and recovery from first episode [23]. Traditionally, based on the type of response shown by the patients, schizophrenia is classified as refractory and non-refractory. Treating refractory schizophrenia (TRS) includes chronic illness and failure to achieve a decline in psychiatric symptoms, despite two adequate treatment trials with antipsychotics from the two different classes (typical and atypical antipsychotics). Refractory patients usually show an appreciable response during the first episode of schizophrenia, predicting the recovery of patients due to the antipsychotics [25].

Earlier studies have indicated that the adverse effects of drugs seriously affect the patient’s adherence behavior [26,27]. They are usually found to be non-responsive to the current medication regime, and almost 34% of schizophrenia patients are reported to have a treatment-resistant schizophrenia [28]. Clozapine is a licensed antipsychotic prescribed for treatment-resistant schizophrenia when there is no adequate response from the repeated use of sufficient doses of at least two separate antipsychotics. Clozapine, though, holds the threat of agranulocytosis and is used in stringent hematological control directives [29]. Moreover, controlled investigations have revealed that electroconvulsive therapy (ECT) is a highly successful therapy for major depression, while being less frequently used in schizophrenia treatment [30]. Several studies have demonstrated the use of drugs alongside ECT to speed up the recovery process [31].

### Aims and Goals

This situation demands a re-evaluation of schizophrenia pathophysiology and the development of novel medication approaches for its treatment and management after reassessing the treatment regime based on the symptoms. Since the challenges associated with the disorder have not been addressed properly, and there is a lack of understanding regarding the trends, prevalence and treatments, there are several gaps that need to be identified. So, the aim of the present study is to determine the frequency of specific diagnostic features that occur in the schizophrenia patients of Pakistan to analyze the effects of different therapies used for the treatment of schizophrenia and the response of the patients to these therapies. The present study is an initiation which has indicated several aspects of schizophrenia management in Pakistan, thus providing us with future directions to identify and rectify the loopholes existing in the current situation.

## 2. Material and Methods

This study was approved by the Ethics committee for the use of human subjects, PMAS-AAUR (No. PMAS-AAUR/IEC/16 and 11 October 2017). It was conducted at University Institute of Biochemistry and Biotechnology, Pir Mehr Ali Shah, Arid Agriculture University, Rawalpindi in collaboration with three psychiatric institutes from Rawalpindi, Lahore and Hyderabad. We collected data from the Benazir Bhutto Hospital, Rawalpindi, Jinnah Hospital, Lahore, Sir Cowasjee Jehangir Institute of Psychiatry, Hyderabad.

### 2.1. Participants and Study Design

Schizophrenia patients from inpatient and outpatient departments of hospitals were assessed and interviewed in accordance with the structured clinical interview for DSM-V (Diagnostic and Statistical of Manual Disorders) [32]. After diagnosis, we selected (N = 200) schizophrenia patients and excluded the patients with an intellectual disability or other organic brain disorders from the study. A consent form was designed and approved by the ethical committee (Ethics Committee for the use of human subjects, PMAS AAUR), which adhered to the Declaration of Helsinki [33]. A detailed questionnaire was developed that was duly filled during the assessment of the schizophrenia subjects.

### 2.2. Assessment Instruments

A data form was used to obtain sociodemographic variables such as age, marital status, age at onset and family history of psychiatric disorder. Family history of the patients was categorized as family history in 1st degree relatives and 2nd degree relatives.

Data that may have been related to age and medication response such as diagnostic parameters and history of drug addiction (tobacco, hashish, marijuana, and cannabis) were also recorded in the information form. On the bases of drug abuse, the data was then classified into three groups: non-addicts, smokers, and multiple drug addicts. Patients using tobacco in any form were classified as smokers. The last group included patients using several drugs such as heroin, hashish, marijuana, cannabis, and tobacco.

### 2.3. Assessment of Therapeutic Approaches

To investigate the treatment potential of the therapeutic approach used for schizophrenia in general, the patients were assessed for their response to these therapies. Then, these patients were categorized into three groups on the basis of the therapy they received. For one group of patients, treatment included all three therapies, viz., medication therapy, psychotherapy, and ECT, and the other group was treated with two therapies: psychotherapy and medication therapy. The other group received only medication and psychotherapy, and third group was treated with just medication therapy. The different medicines used for the medication therapy included typical anti-psychotics, atypical anti-psychotics, EPS inhibitors, anti-depressants, anti-manic, anti-dementia, anti-epileptics, and high-potential depot injections. Since the depot injection is injected intramuscularly, they were given to the patients who had aggressive behavior—most of them were not willing to take oral medication. These patients did not show any response to oral medication either. This assessment period spanned 2 to 4 weeks. The data was then recorded after the detailed analysis of the patients to record the response. On the bases of this observation, the schizophrenia subjects were classified as treatment resistant and non-resistant. Resistant and non-resistant schizophrenic patients assessed for the medication response were further classified into separate groups: 1—patients with no response (N-R), 2—temporary slow response (S-R), 3—response, followed by the worst episodic attacks, 4—completely recovered (C-R).

### 2.4. Statistical Analysis

The finalized data was then compiled and analyzed by R-4.1.0. Shapiro–Wilko test was applied to check the distribution of data. The relationships between the different variables were found by correlation analysis, and a correlation matrix was created. The frequencies of the different diagnostic and sociodemographic parameters were calculated. The variables with Pearson correlation coefficient values r ≥ 6 were further subjected to linear regression analysis to find an association between these variables.

The variation between the patient’s responses to treatment therapy was calculated using ANOVA, and variation within different treatment responder groups was calculated by post-hoc Tukey HSD test.

## 3. Results

A total of 200 diagnosed schizophrenia cases were identified from the different psychiatric organizations, as indicated in the methods. Among the 200 schizophrenia subjects, 73.5% (*n* = 147) were male subjects and 26.5% (*n* = 53) were females. About 50.5% (N = 101) of the cases belonged to the first age group, in which the disorder had developed between 15 and 25 years of age. In these cases, 1% (N = 2) of the patients were observed to have some motor dysfunction/psychiatric symptoms in their childhood, wherein after adolescence they were diagnosed with schizophrenia. Calculated frequencies of the disorder for the different age groups demonstrated that the age group of 26–35 years of age included about 10.5% (N = 21) of the patients, and 6% (N = 12) of the cases had onset between 36 and 50 years of age, with the minimum age of onset in childhood (8 years). The age of the schizophrenia patients in this randomly collected samples/group ranged from 16 to 68 years (mean ± SD = 34 ± 10). The region-wise distribution of patients in the studied sample indicated the comparative frequency of different ethnic groups: N = 85 for Punjabi, N = 31 for Kashmiri, N = 22 for Pakhtoons, and N = 16 for other small ethnic groups. The ethnicity of 46 individuals was not disclosed (Table 1).

### 3.1. Sociodemographic and Diagnostic Characteristics

The overall calculated frequency for the different clinical features of schizophrenia in the present study was learning disability—76% (*n* = 151), poor social and occupational functioning—75.5% (*n* = 151), hallucinations—75% (*n* = 150), drug abuse—18.5% (*n* = 37), fear—19.5% (*n* = 39), paranoia—20.5% (*n* = 41) and stressor—10% (*n* = 20) (Figure 1). Among the 200 schizophrenia subjects, the overall frequency of non-responders was 5% (*n* = 10), the frequency of patients in which the response was followed by the worst episodic attacks was 24% (*n* = 48), temporary slow responders were 42% (*n* = 86) and that of the completely recovered patients were 1.5% (*n* = 3). In the studied group of schizophrenia patients, psychotherapy in combination with medication therapy was used for 36.5% (*n* = 73) patients (Figure 2). Of these 73 patients, 20% (*n* = 15) were found to be non-responders and 45.2% (*n* = 33) subjects were reported to show immediate response, followed by the worst episodic attack. The slow responders were 28% (*n* = 20) of the studied subjects. The schizophrenia patients who were subjected to ECT showed a short-term response (patients were not mentally stable with the severity of symptoms) (Figure 1b). The response of 41.2% (*n*= 30) of the patients was uncertain. Calculated frequencies of the prescribed drugs indicated that typical and atypical antipsychotics were used more frequently than all other drugs (Figure 3).

### 3.2. Correlation between Different Diagnostic and Sociodemographic Variables

Correlation analysis indicated that there was a significant correlation between some of the parameters (Figure 4). Variables which show a correlation coefficient r ≥ 0.6 in the correlation matrix are considered significantly correlated. The correlation matrix (Appendix A) indicated that auditory hallucination was significantly correlated with paranoid symptoms (r = 0.638), ECT treatment (r = 0.65), no. of ECTs (r = 0.688), underlying medical condition (r = 0.69), fear (r0.58) and depot injection treatment (r = 0.616). Stress, as a trigger of psychosis, was found to be strongly correlated with fear (r = 0.61) and paranoia (r = 0.77). Significant correlation values were found between fear and mood swings (r = 0.67) and disorganized behavior (0.64). A significant correlation is observed between mood swings and drug addiction (r = 0.67), disorganized behavior (r = 0.655), paranoia (r = 0.637), ECT (r = 0.634) and no. of ECTs (0.68). Drug addiction in schizophrenia has a significant correlation with learning disability (r = 0.60), ECT (r = 0.65 and no. of ECTs (r = 0.65). Disorganized behavior is associated with higher correlation values to learning disability (r = 0.7), no. of ECTs (r = 0.71), underlying medical condition (r = 0.716) and paranoia (r = 0.61). High correlation values are found between drug addiction and learning disability (r = 0.60), ECT (r = 0.65), and no. of ECTs (0.655).

The variables that have significant correlation were subjected to a generalized linear regression model (Table 2), which showed that patients’ response to medication were significantly associated with the age, gender, age at onset, and marital status (at 95% C. I), while medication response was not associated with ECT treatment, drug addiction, and ethnicity. Regarding age at onset, the *p*-value (0.00864) shows that the association is significant, but the CI range (0.839 and 1.0008) and OR (0.916) values reflecting that effect are small. Moreover, the CI range is narrow, which further supports the significance of the results. The schizophrenia cases with both first-degree family history and second-degree family history were included in a separate category of “familial cases”. The multivariate regression model applied between familial case and other parameters indicated that not a single variable (age at onset, drug addiction, response to medication, stress/trigger, previous psychiatric hospitalization, age, gender, learning disability, paranoia) was associated with family history (Table 3).

### 3.3. Analysis of Variance in Treatment Responses

The variation among treatment responses was investigated by using an ANOVA, which showed an association with the typical antipsychotics (*p* = 0.00128) at a 95% family wise confidence interval. A post-hoc Tukey analysis further identified the differences between groups (Figure 5), which showed that a significant difference lies (*p*-adj = 0.003) between group 3 (episodic response followed by worst episodic attacks) and group 4 (completely recovered). Similarly, a significant association exists between the medication response and the combination treatment (ECT and antipsychotics) (*p* = 0.00258, F = 4.981). The Post-hoc Tukey HSD analysis further identified a difference within the medication response categories. A significant difference exists between group 3 (episodic response followed by worst episodic attacks) and group 4 (completely recovered) (*p*-adj = 0.006), and no significant variation among the treatment responses was observed with other prescribed drugs, which include atypical antipsychotics, anti-manic, antidepressants, anxiolytics, and depot injection (Figure 6).

## 4. Discussion

This study adds to the previous knowledge regarding schizophrenia diagnosis and treatment and its association with multiple factors. We found that schizophrenia-specific characteristics occurred with the same frequency in the studied sample compared to in previous studies, and that the differences between genders as a temperament trait to predict the disorder are not significant. The comparison of the frequencies of diagnostic and sociodemographic features on the basis of their significance in diagnosis after the detailed assessment suggests that these features occurred with the same frequency in our sample as was observed in the other populations [34,35]. Regarding age, in 50.5% of the patients, the psychotic symptoms appeared in their teenage years. Past studies also indicated that it develops in the teenage years or soon after adolescence, but most of the time it is concurrent with other neurological and psychiatric disorders [4,36,37]. There is evidence in the literature that supports the transformation of neurological disorders in childhood to schizophrenia or other mental disorders such as autism spectrum disorder in adulthood [38]. We have also reported a single case where the onset of schizophrenia occurred in childhood (4 years), which supports the idea of the transformation or progression of childhood neurological disorders to schizophrenia at adult age, however further research on such rare cases is required to have a clear understanding of the incidence of such a case.

Regarding the patients’ response to treatment therapies in different age groups, the results indicate that that patient response is strongly associated with the age of the patient and patient’s age at the onset of the disorder (for age, *p* = 0.021, OR = 0.07 and for age at onset, *p* = 0.0086, OR = 0.916). This suggests that early onset schizophrenia patients are more impaired and clinically severe than those where onset occurs later in life [39]. Early onset schizophrenia may be a consequence of a distinct neurobiological entity compared to that of adult onset, which supports the neurodevelopmental hypothesis of schizophrenia. After the first acute episode, the prompt initiation of drug treatment is vital, and appropriate dosing is titrated based on the patient’s response [40]. This study indicated that patients treated with both the augmentation and combination therapy were more stable compared to those treated with only antipsychotic agents. The results indicate that the response to medication is affected by the type of treatment regime followed. In the studied group, typical antipsychotics were shown to significantly influence the response of the patients. Moreover, significant variation among treatment responses was observed due to the antipsychotic agents. In the studied sample, antipsychotics were observed to produce long-term effects in most patients with schizophrenia. Past studies have also indicated that the use of both augmentation and combination therapy has a significant advantage in terms of rapidity as well as quality of response [41,42,43,44,45,46,47]. Evidence from the sample studied with respect to the response to ECT suggests that its short-term effects are followed by severe episodic attacks as compared to the stable nonaggressive behavior of the patients treated with the medication therapy in combination with the psychotherapy. Some studies suggest that the patients with schizophrenia who stop taking medication are at increased risk of relapse, which can lead to hospitalization [48]. Analyses have revealed that the patients receiving drugs at a younger age respond relatively better than the patients who are older at the time of prescription. Patients treated with drugs have shown a positive and more persistent response compared to those treated with frequent ECTs. A positive response to ECT was observed in the studied sample, but it was very short term and was followed by aggression and severe episodic attacks—hence why most of the psychiatric organizations in Pakistan rarely use ECT.

We found significant associations of medication response with marital status (*p* = 0.005, OR = 6.18) and gender (*p* = 0.0016, OR = 5.28). Regarding gender differences in antipsychotic treatment response, most studies with typical antipsychotics have also indicated that women with schizophrenia show a fast and better response to typical antipsychotics than men [2]. Marital status seems to influence the behavior of the schizophrenia patient as well. A longitudinal study based in China supports the fact that marriage can be instrumental in improving family-based support and care, as well as improving the community tenure of people with schizophrenia [49]. Our study also indicates that the age of onset, it is associated with age, but the OR is small; however, here, we have two measures of significance. Though the *p*-value is significant, the smaller OR and CI values closer to 0 or crossing 0 are telling that the effect is small, despite the *p*-values reflecting association. Moreover, there is evidence that shows that, regarding a narrow CI reflecting significance of results versus a wider CI, as general rule, “The narrower the CI, better it is”. Therefore, it is important to develop programs to enhance the opportunity for persons with schizophrenia to get and stay married [49].

Antipsychotic agents in combination with other drugs are used to treat symptoms such as hallucinations, fear and aggression, but in addition to focusing on patients, treatment programs that encourage family support have been shown to decrease rehospitalization and improve social functioning [42]. Social withdrawal, among other abnormal behavior, typically proceeds a person’s first psychotic episode, but sometimes individuals do not exhibit all of the characteristic symptoms. Our study indicates hallucinations, poor social functioning and a learning disability as the most frequent symptoms that schizophrenia patients are observed to exhibit at onset. As states earlier, it is characterized by positive, negative and cognitive symptoms, each of which are vitally important to the clinical attempts to distinguish schizophrenia from other mental disorders. Positive symptoms are easily diagnosed, but it is difficult to identify negative symptoms. The treatment options available target these symptoms and act to lower or diminish the symptoms. The primary symptoms and comorbid conditions associated with schizophrenia may ultimately lead to social and occupational dysfunction. The onset of psychotic symptoms at an early age and the treatment response are linked to each other, but non response to these treatment therapies indicates that the symptom severity involves a complex neurobiological background, and the patients show resistance to treatment. Treatment-resistant schizophrenia may be different from treatment responsive schizophrenia, which suggests that there might be more neurodevelopmentally oriented pathophysiological processes that could be involved. The data also suggests that there could be the common neurobiological backgrounds that associate the treatment response of schizophrenia with the age at onset. Moreover, the non-responders were more clinically severe and impaired than other schizophrenia patients, which is also a condition of non-response to antipsychotics, which can be regarded as a predictor of a more severe neurobiological type of schizophrenia.

The detailed data sets describing the family history of schizophrenia showed that individuals with a family history of any neuropsychiatric disorder are more susceptible to schizophrenia. It was also observed that most of the patients with diagnosed schizophrenia had first-degree relatives with a psychiatric disorder. This indicates that it is a heritable disease with a strong association of its occurrence with mental disorders in first-degree relatives [5]. A novelty in this study lies in the studied population. There have been no prior studies conducted in Pakistan to investigate the treatment responses of schizophrenia patients. The whole treatment regime currently being followed is based on the research conducted on European populations. Therefore, this was the beginning of a new area of research that will hopefully be followed by studies that will expand the understanding of the underlying cause/molecular mechanisms of schizophrenia in Pakistan.

### Limitations

The sample size could be the limitation of this study.

## 5. Conclusions

The current study indicates a large number of aspects that are related to psychiatric health that are yet to be investigated. Schizophrenia is a complex disorder that requires prompt treatment at the first sign of a psychotic episode. To develop a comprehensive treatment plan, clinicians must consider treatment-related adverse effects and the potential for nonadherence as well. Although we cannot rule out that the subset of patients responded differently to treatment, overall, the small differences in outcome variance suggests that average treatment effect might be the reasonable assumption for an individual patient.

## Figures and Tables

**Figure 1 brainsci-13-00407-f001:**
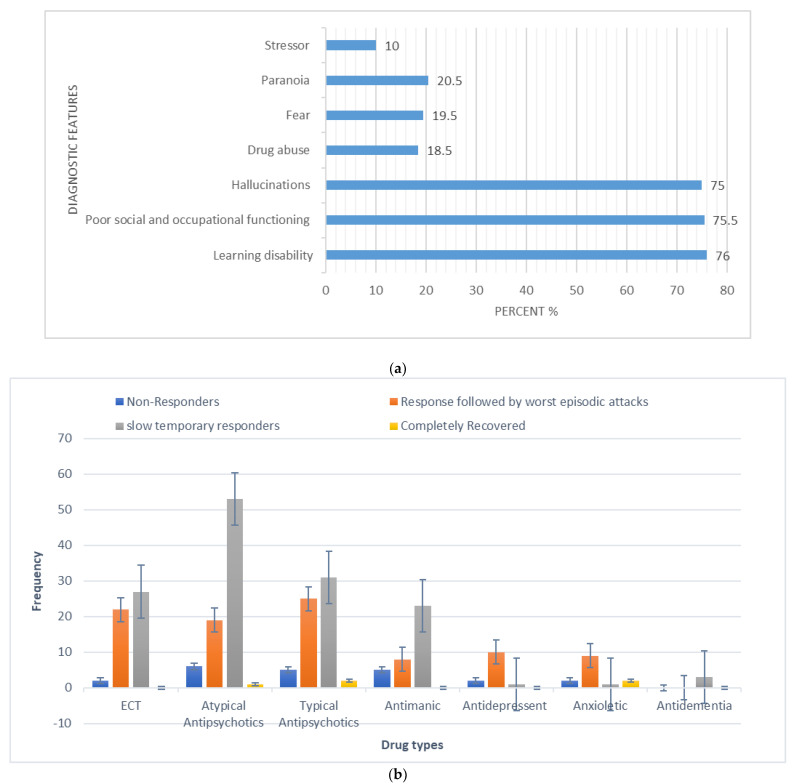
(**a**) Frequency (%) of diagnostic (core) features in the studied schizophrenia sample (N = 200). (**b**) Categorization of responses to polypharmacy treatments given to schizophrenia patients in the studied sample (X-axis represents the different types of drugs and Y-axis represents the frequency of occurrence).

**Figure 2 brainsci-13-00407-f002:**
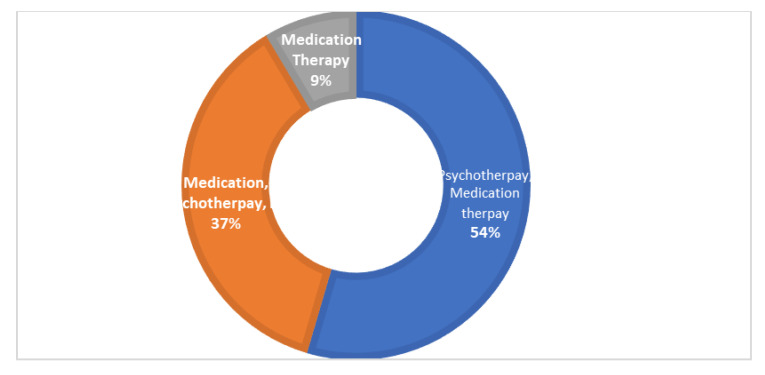
The pie donut chart represents different types of therapies used to treat the studied group of patients. Psychotherapy with medication therapy was given to 54% of the patients, medication therapy, psychotherapy and ECT combined were given to 37% of patients and medication therapy alone was given to 9%of the patients.

**Figure 3 brainsci-13-00407-f003:**
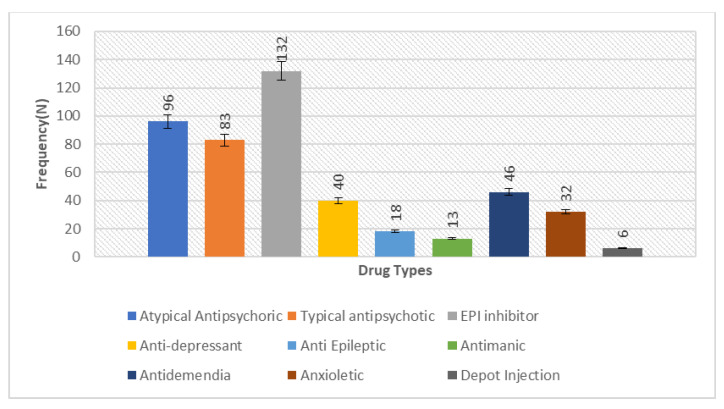
Bar chart depicting the measure of prescription frequency of drugs in the studied sample of 200 schizophrenia patients, representing the comparative higher frequency of antipsychotics (typical 96 and atypical 83) to lower the severity of symptoms in the disorder.

**Figure 4 brainsci-13-00407-f004:**
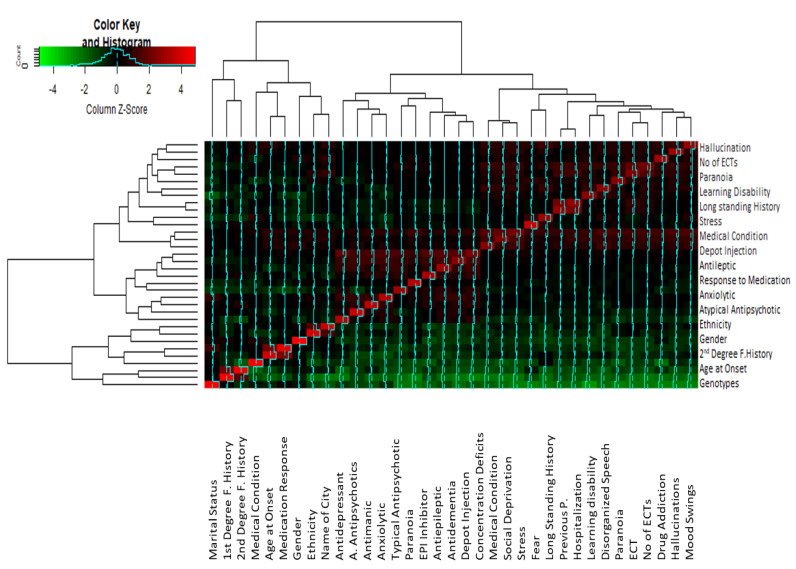
Cluster heatmap showing the correlation between different studied variables. The clusters analysis links the different values on the base of the z-score differences, and the correlated variables are clustered together with the least difference between them. The closely related variables with maximum z-scores are shown in red, while those with minimum z-scores are represented by green. Note: F. History refers to Family history. Antipsychotics refers toatypical antipsychotics.

**Figure 5 brainsci-13-00407-f005:**
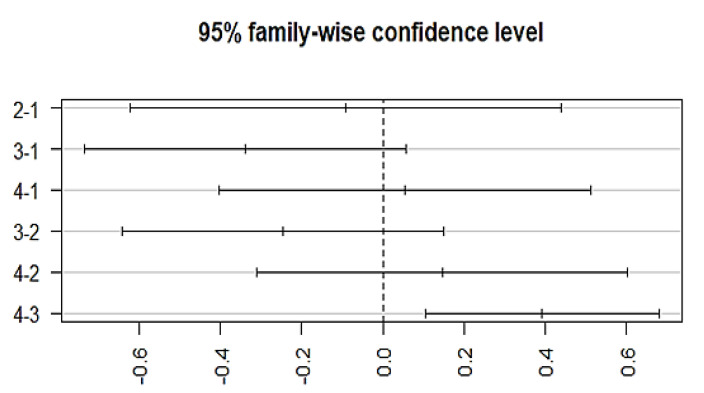
Posthoc analysis representing the differences in the mean levels of response to medication by typical antipsychotics. X-axis indicates the categorization of different groups based on the medication response. (Group-1) No response, (Group-2) temporary slow response, (Group-3) response followed by the worst episodic attacks, (Group-4) completely recovered. (2-1)Difference in response to medication between group 2 and 1, (3-1) difference in response to medication between group 3 and 1, (4-1) difference in response to medication between group 4 and 1, (3-2) difference in response to medication between group 3 and 2, (4-2) difference in response to medication between group 4 and 2, and (4-3) difference in response to medication between group 4 and 3.

**Figure 6 brainsci-13-00407-f006:**
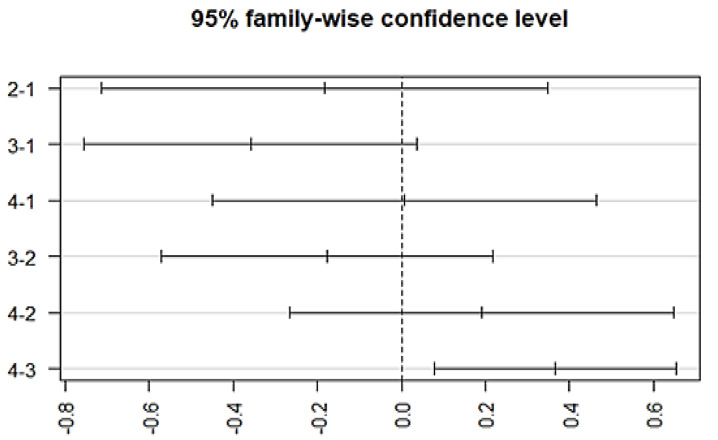
Post-hoc analysis representing differences in the mean levels of medication response regarding combined treatment therapy (ECT combined with antipsychotics). (Group-1) No response, (Group-2) temporary slow response, (Group-3) response followed by the worst episodic attacks, (Group-4) completely recovered. (2-1) Difference in response to medication between group 2 and 1, (3-1) difference in response to medication between group 3 and 1, (4-1) difference in response to medication between group 4 and 1, (3-2) difference in response to medication between group 3 and 2, (4-2) difference in response to medication between group 4 and 2, and (4-3) difference in response to medication between group 4 and 3.

**Table 1 brainsci-13-00407-t001:** Summary of all sociodemographic parameters studied for schizophrenia group.

Gender (N)		Age	Age at Onset (y)				Ethnicity (%)				
Male	Female	Mean ± SD	15–25	26–35	36–50	Unknown	Punjabi	Kashmiri	Pakhtoons	Other Ethnic Groups	Unknown
147	53	34 ± 10	50.5%	10.5%	6%	33%	42.5%	15.5%	11%	8%	22%

**Table 2 brainsci-13-00407-t002:** Association between medication response and other variables in the generalized linear regression model.

Variables	OR (95% Confidence Interval)	*p*-Value
Age	0.074 (0.0082–0.680)	0.0231
Age at onset	0.916 (0.839–1.0008)	0.00864
Gender	6.183 (1.695–22.544)	0.00577
ECT	1.672 (0.201–13.901)	0.634
Drug Addiction	1.407 (0.166–11.905)	0.7537
Ethnicity	0.422 (0.1048–1.703)	0.225
Marital Status	5.28 (1.4921–18.683)	0.00169

**Table 3 brainsci-13-00407-t003:** Multivariate regression model to show association between family history of schizophrenia and sociodemographic parameters.

Variable	B	S. E	T	*p*
Age of Onset	−1.190× 10^−17^	1.519 × 10^−17^	−7.840 × 10^−1^	0.435
Drug Addiction	−2.156 × 10^−17^	2.017 × 10^−16^	−1.070 × 10^−1^	0.915
Response to medication	9.182 × 10^−17^	1.476 × 10^−16^	6.220 × 10^−1^	0.536
Stress	1.214 × 10^−16^	2.439 × 10^−16^	4.980 × 10^−1^	0.620
Hospitalization	1.041 × 10^−16^	2.106 × 10^−16^	4.940 × 10^−1^	0.622
AGE	−1.539 × 10^−18^	6.355 × 10^−18^	−2.420 × 10^−1^	0.809
Gender	2.762 × 10^−17^	1.128 × 10^−16^	2.450 × 10^−1^	0.807
Learning Disability	−2.340 × 10^−17^	9.725 × 10^−17^	−2.410 × 10^−1^	0.810
Paranoia	−8.181 × 10^−17^	1.177 × 10^−16^	−6.950 × 10^−1^	0.489

Note. S.E (standard error) value is for unstandardized beta (B); *p*-value shows the significance of association between the variables in C. I = 95%.

## Data Availability

The data presented of this study are available on request from Corresponding authors. The data are not publicly available due to confidentiality of patient’s information.

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
