# Peer review of "Differential Treatment Responses in Pakistani Schizophrenia Samples: Correlation with Sociodemographic Parameters, Drug Addiction, Attitude to the Treatment and Antipsychotic Agents"

_brainsci, 2023, doi:10.3390/brainsci13030407_

Round 1

Reviewer 1 Report

This is a paper focused on patients with schizophrenia and their response to antipsychotic treatments. The topic of the paper is of interest for the journal; however, several major changes are recommended before considering it for publication.

Abstract.

1- The introduction of the abstract is really confusing. What are the main aims of the paper? To describe antipsychotic response or to find factors potentially influencing antipsychotics?

2-The patients are defining different "patterns" of antipsychotic response. This is really confusing. Are they defining response qualitatively or quantitatively? Non-response vs. slow response? 

3- Please, define non-response and response. Is it a subjective definition?

4-The conclusion of the abstract is really long. Please, rebuilt it and summarize conclusions.

Introduction

1-The introduction is mainly based on schizophrenia, being a general introduction provided. I recommend to expand the section about response to treatments. What about explaining definition of response, refractory and non-refractory patients.

2-Please provide aims and goals in a separate subsection.

Materials and methods.

1-The first part should be renamed as "Participants and study design". 

2-The section section should be assessment instruments.

Results

1- Figure 1 difficult to understand. Please provide diagnoses according to the current diagnostic and statistical manuals.

Discussion

What is new in the findings the authors are reported? It is difficult to identify new findings in the treatment of schizophrenia? 

Are men and women responding similar or differently? this should be better explained...

Author Response

We thank the reviewer for a careful review and constructive comments on our manuscript. The point-by-point response to each comment is provided below. 

1-The introduction of the abstract is really confusing. What are the main aims of the paper? To describe antipsychotic response or to find factors potentially influencing antipsychotics?

ANS: We apologize for this. Now we have made the abstract clear and the aim of the study is spelled out. (page#1, line 21-23)

2-The patients are defining different "patterns" of antipsychotic response. This is really confusing. Are they defining response qualitatively or quantitatively? Non-response vs. slow response? 

ANS: This response is qualitative as the final response was recorded by psychiatrists after complete assessment of patients. On the bases of this assessment, the patients were grouped into four types mentioned in the manuscript.

3- Please, define non-response and response. Is it a subjective definition?

ANS: Yes, it is a subjective term.

Non responders is the group of patients who did not show any appreciable response to medication therapy within the assessment period. We have added/stated on page #4, line #170-171)

4-The conclusion of the abstract is really long. Please, rebuilt it and summarize conclusions.

ANS: The precise conclusion is added in manuscript on page#1, line#37-40

“The data suggests proper monitoring of patient’s behaviour to treatment therapies to implement tailored interventions. Despite several genetic studies supporting the heritability of schizophrenia, insignificant association between characteristic features and family history might be due to limited sample size, suggesting to work collaboratively with massive sample sizes.”

Introduction

1-The introduction is mainly based on schizophrenia, being a general introduction provided. I recommend to expand the section about response to treatments. What about explaining definition of response, refractory and non-refractory patients.

ANS: Details of the treatment response are added in the revised manuscript.  please see page #2, line 92-108.

2-Please provide aims and goals in a separate subsection.

ANS: Aims and goals are mentioned in separate section. Please see page 3, line 119-129

Materials and methods.

1-The first part should be renamed as "Participants and study design". 

ANS: As suggested, we have renamed it in the revised version of the manuscript please, section 2.1, page # 3

2-The section should be assessment instruments.

ANS: As suggested, we have renamed it in the revised version of the manuscript please, section 2.2, page # 3

Results

1- Figure 1 difficult to understand. Please provide diagnoses according to the current diagnostic and statistical manuals.

ANS: The figure 1 (pie graph) is replaced by bar graph which is explaining the frequency with which the characteristic features of schizophrenia are found in the studied group of schizophrenia. The criterion of diagnosis was traditionally same therefore it is not mentioned in separately. We have calculated the frequency of features to analyse which features most commonly exist in schizophrenia patients of Pakistan.

These patients included in the study were diagnosed by the psychiatrists according to (DSM IV) as mentioned in the methods section:

“2.1 Participants and study design

Schizophrenia patients from inpatient and outpatient department of hospitals were assessed and interviewed in accordance with the structured clinical interview for DSM-IV [33].

Discussion

What is new in the findings the authors are reported? It is difficult to identify new findings in the treatment of schizophrenia? 

ANS: The novelty in the study lies in the studied population. The study is conducted to find the variation in the treatment response of schizophrenia patients in Pakistan. There are no prior studies conducted to investigate the treatment responses of the schizophrenia patients based in Pakistan. The whole treatment regime being followed is based on the researches conducted on European populations. Therefore, it is an initiation of such type of research and, will be followed by studies to understand the underlying cause/ molecular mechanisms.

It is stated in the manuscript. Please see (page 10, line 369-375)

Are men and women responding similar or differently? this should be better explained...

ANS: Since the number of females in sample is relatively less than the males that’s why the proportion test does not identify precise difference in response of females and males. In addition the response is divided into groups and results of proportion test seems biased due to smaller sample size.

Reviewer 2 Report

Dear Editor,
I really appreciate the opportunity to review the manuscript brainsci-2187552 entitled:
"Differential treatment responses in Pakistani schizophrenia sample: Correlation with Sociodemographic parameters, Drug addiction, attitude to the treatment and antipsychotic agents"

I commend the authors for describing this critical and timely issue. The paper is interesting and well-written; however, I would like to highlight some issues that merit revision:

The study presented appears to be absolutely excellent in predisposition and presentation, and I have only one question to ask the authors, regarding any psychotherapeutic treatment toward the patients. It may have been missed, but if not, I beg the authors to specify in the text whether, and if so what kind and in what percentage the patients benefited from psychotherapy, in add-on to drug treatment; if so if the data is not available I beg the authors to add it, even with one line, between the limitations. Excellent work, I congratulate the authors once again

Author Response

We thank the reviewer for careful review of our manuscript, and the comments. 

The study presented appears to be absolutely excellent in predisposition and presentation, and I have only one question to ask the authors, regarding any psychotherapeutic treatment toward the patients. It may have been missed, but if not, I beg the authors to specify in the text whether, and if so what kind and in what percentage the patients benefited from psychotherapy, in add-on to drug treatment; if so if the data is not available I beg the authors to add it, even with one line, between the limitations. Excellent work, I congratulate the authors once again

ANS: Since schizophrenia is debilitating mental disorder and involves the chemical imbalance that’s why the treatment regime being followed includes use of drugs (antipsychotic agents) plus psychotherapy and sometimes ECT as well. Psychotherapy is never used alone for the treatment of schizophrenia. The percentage of patients which were treated with medication therapy combined with psychotherapy and their response is mentioned in the text. Please see (page#4, line#205-209).

Reviewer 3 Report

In this work, the authors perform a series of statistical correlation analyses to stablish the relationship between several features of the patients and the clinical picture, focusing the analysis on the response to the treatment. Despite the fact that this kind of analyses have been previously published (see Hofer et al., 2005 or Xiang et al., 2008 as examples), it is interesting to consider these relationships in different countries and cultures due to the fact that environmental factors are quite important in the etiology and development of schizophrenia.

However, in my opinion the paper must be completely reformulated before considering its publication:

GENERAL CONCEPTS

The manuscript must be re-read and English corrected. There are numerous misprints and incorrections, including words that are not separated (such as, “ isevidence” in line 275 or “includingSchizophreniapatients” in line 20), typos (like “antipsychoric” or “ansioletic” in figure 3), incorrect use of capital letters (use of capital letters in “Schizophrenia” throughout the text) or anacoluthons (“The study adds to the previous knowledge about the schizophrenia diagnosis and treatment and its association with multiple factors” in lines 264-265). Also, in the abstract different fonts are employed.

TITLE

The authors mention the “attitude to the treatment” in the title of the publication (line 2), but it is not considered in the text anymore.

ABSTRACT

 The meaning of the first sentence of the abstract is not clear: “There is an assumption in clinicians and researchers that schizophrenia patients vary considerably to different treatments therapies including augmentation therapy (ECT or a mood stabilizer) and the combination therapy (with antipsychotics)” (line 14-16).

If the authors choose to include statistical analysis in the abstract (i.e. “p=0.00258” in line 31), they should indicate the employed test. Also, usually the statistical significance is provided as relative values (p<0.01), and the asterisks are not usually included in the abstract.

INTRODUCTION

The authors must consider cognitive symptoms in schizophrenia apart from positive and negative ones (line 51-54) (Carbon and Correll, 2014).

The authors should differentiate between typical and atypical antipsychotics in lines 59-80 in the introduction because they have different mechanisms of action, side effects and symptoms improved. 

The authors should mention the role of drug addition and other risk factors of schizophrenia in the introduction because they consider them latter in the results section and in the discussion.

In line 82, the authors cite a paper by Locke et al. (2015). That paper is referenced in the bibliography (lines 417-583), but its topic is the relationship between genetic features and BMI. The authors must confirm if it is correct.

METHODS

The expression “mental retardation” sounds too rough nowadays (line 106). Please, use “intellectually disabled”.

The authors must say which ethical committee approved the study (line 107).

Tobacco is included both in the smokers’ group and in the multiple drug addicts’ group. Is it correct? (line 121-122)

The authors consider three groups of patients: medication therapy + psychotherapy + ECT, medication + psychotherapy and only medication (lines 126-130), but I cannot see any analysis done considering those three groups. Also, how many patients are there in each group?

Which drugs are considered between the “anti-dementia”? (line 132) Sometimes, antipsychotics are employed in dementia, such as risperidone or quetiapine (Yunusa et al., 2019). Also, in schizophrenia most of the “depot injections” (line 133) employed are antipsychotics. Authors must take it in consideration.

What do the authors exactly mean with “EPI inhibitors”? (line 132).

Which correlation tests are employed (line 145), Pearson test or other?

Authors should consider the assessment of the symptomatology of the patients (prevalence of positive and negative symptoms, cognitive tests…).

RESULTS

A table to resume the features of each group of patients (lines 151-163) would be helpful to interpret the results.

A pie chart (figure 1) may not be the most adequate graph to represent the frequency of diagnostic symptoms because some of them could appear at the same time in the same patient. Also, in the text other percentages are mentioned (lines 167-170).

In figure 2, in the Y-axis authors should indicate that it is a frequency measurement (as you do in figure 3).

In line 200, authors cite a Supplementary table 1 that is not provided (it was confirmed by the editor). It would be vey helpful to see the numeric values of the correlation tests.

In table 1, authors say that the association between medication response and age of onset has an Odds Ratio between 0.839 and 1.0008. With those values, I find difficult that the p value is <0.01.

In table 2, I cannot see the purpose of considering the association of the gender with the family history of schizophrenia.

Which gender are the authors considering in figure 4, table 1, table 2…? (male or female).

In figures 5 and 6 the authors perform a group analysis (2-1, 3-1…). It must be explained which is the meaning of those codes.

DISCUSSION

“There isevidence (SIC) in literature that support the transformation of neurological disorders in the childhood to Schizophrenia or other mental disorders as autism spectrum disorders at adult age” (lines 275-277). That asseveration must be correctly referenced.

All through the discussion, authors include results that were not exposed in the results section. For example, “Regarding age, in 60% of the patients the psychotic symptoms appeared before 19 years of age” (lines 271-273).

Also, in some parts of the discussion authors take speculative conclusions from the results. For example, “Early onset schizophrenia may be a consequence of distinct neurobiological entity compared to adult onset which support the neurodevelopmental hypothesis of schizophrenia” (lines 286-288).

There are results that are not discussed in the discussion section. For example, the cluster correlation analysis (lines 198-213) or the analysis of treatment responses (lines 245-256).

CONCLUSIONS

“Although we cannot rule out that subset of patients responds differently to treatment, overall small differences in outcome variance suggests that average treatment effect might be the reasonable assumption for individual patient” (lines 340-342). The meaning of that sentence is not clear.

REFERENCES

Carbon M, Correll CU. Thinking and acting beyond the positive: the role of the cognitive and negative symptoms in schizophrenia. CNS Spectr. 2014;19(S1):35–53.

Hofer A, Baumgartner S, Edlinger M, Hummer M, Kemmler G, Rettenbacher MA, et al. Patient outcomes in schizophrenia I: correlates with sociodemographic variables, psychopathology, and side effects. Eur psychiatr. 2005;20(5-6):386-94.

Xiang Y, Ma X, Cai Z, Li S, Xiang Y, Guo H, et al. Prevalence and socio-demographic correlates of schizophrenia in Beijing, China. Schizophrenia Research. 2008;102(1-3):270-7.

Yunusa I, Alsumali A, Garba AE, Regestein QR, Eguale T. Assessment of Reported Comparative Effectiveness and Safety of Atypical Antipsychotics in the Treatment of Behavioral and Psychological Symptoms of Dementia: A Network Meta-analysis. JAMA Netw Open. 2019;2(3):e190828.

Author Response

We thank the reviewer for a careful review and constructive comments on our manuscript. The point-by-point response to each comment is attached. 

Round 2

Reviewer 1 Report

The authors have substantially improved the paper by following the reviewers' recommendations. I consider that the paper can be accepted.

Author Response

Response to reviewer 1

Manuscript is language- and grammar-checked by the experts.

The changes have been tracked and updated draft is provided.

Introduction was changed in the last review and all the missing information was added, supported by the references

Reviewer 3 Report

The authors provide detailed responses to all the formulated questions. Nevertheless, language must be checked and corrected before the publication of the paper. There are still several misprints, for example, “currentstudy” (line 21), “disabilityor” (line 140), “antidepressent” (figure 1b), “variationamong” (line 280), among others. Also, please confirm that in line 177, when you say “r>6”, it is correct.

Author Response

Response to reviewer 3

Review-             The authors provide detailed responses to all the formulated questions. Nevertheless, language must be checked and corrected before the publication of the paper. There are still several misprints, for example, “currentstudy” (line 21), “disabilityor” (line 140), “antidepressent” (figure 1b), “variationamong” (line 280), among others. Also, please confirm that in line 177, when you say “r>6”, it is correct.

Response-       

  • Whole document is language checked. The updated draft with the tracked changes is uploaded.
  • Misprints are corrected.
  • It is r≥6. Yes, it is correct.
